# NiTi SMA Superelastic Micro Cables: Thermomechanical Behavior and Fatigue Life under Dynamic Loadings

**DOI:** 10.3390/s22208045

**Published:** 2022-10-21

**Authors:** Paulo C. S. Silva, Estephanie N. D. Grassi, Carlos J. Araújo, João M. P. Q. Delgado, Antonio G. B. Lima

**Affiliations:** 1Multidisciplinary Laboratory of Active Materials and Structures (LaMMEA), Department of Mechanical Engineering, Federal University of Campina Grande, Campina Grande 58429-140, Brazil; 2CONSTRUCT-LFC, Civil Engineering Department, Faculty of Engineering, University of Porto, 4200-465 Porto, Portugal

**Keywords:** shape memory alloys, superelasticity, NiTi SMA micro cable, self-heating, fatigue life

## Abstract

Shape memory alloy (SMA) micro cables have a wide potential for attenuation of vibrations and structural health monitoring due to energy dissipation. This work evaluates the effect of SMA thermomechanical coupling during dynamic cycling and the fatigue life of NiTi SMA micro cables submitted to tensile loadings at frequencies from 0.25 Hz to 10 Hz. The thermomechanical coupling was characterized using a previously developed methodology that identifies the self-heating frequency. When dynamically loaded above this frequency, the micro cable response is dominated by the self-heating, stiffening significantly during cycling. Once above the self-heating frequency, structural and functional fatigues of the micro cable were evaluated as a function of the loading frequency for the failure of each individual wire. All tests were performed on a single wire with equal cross-section area for comparison purposes. We observed that the micro cable’s functional properties regarding energy dissipation capacity decreased throughout the cycles with increasing frequency. Due to the additional friction between the filaments of the micro cable, this dissipation capacity is superior to that of the single wire. Although its fatigue life is shorter, its delayed failure compared to a single wire makes it a more reliable sensor for structural health monitoring.

## 1. Introduction

Shape memory alloys (SMA) based on nickel–titanium (Ni-Ti) have wide application potential in dynamic regimes due to the functionality of superelasticity, being capable of recovering high levels of deformation, of the order of up to 10% in tension, after mechanical unloading [1]. Moreover, the mechanical hysteresis generated during the isothermal loading/unloading cycle is capable of dissipating mechanical energy. These phenomena occur due to reversible solid-state phase transformations between two crystallographic structures, austenite and martensite. As the phase transformation that originates the superelasticity in SMA is intimately related to temperature, so is their mechanical behavior. Ultimately, the stresses that trigger the forward (loading) and reverse (unloading) phase transformations increase linearly with temperature, following the Clausius–Clapeyron law [2,3].

The stress-induced phase transformations are associated with the release and absorption of latent heat, being exothermic during the forward transformation (austenite to martensite) and endothermic during the reverse transformation (martensite to austenite). At low strain rates, the heat generated during these phase transformations is released to the medium, allowing maintenance of a thermal balance. However, in dynamic regime applications where repetitive superelasticity cycles are required under loading frequencies sufficiently high (which mostly depends on the surface/volume ratio of the SMA element), this thermal equilibrium is affected and the material experiences a gradual accumulation of heat, which consequently leads to an increase in the internal temperature due to the Clausius–Clapeyron law. This thermomechanical coupling is the cause of the strain rate dependence observed in SMA [4,5,6]. As a result, the hysteretic loop generated by the difference between the load/unload paths during a superelastic cycle decreases in size, affecting the energy dissipation capacity of the material [7]. Therefore, it is crucial for the designer to investigate the thermomechanical behavior of SMA structural elements under a dynamic regime, according to the system prerequisites.

The cable geometry, formed by several bulk elements twisted around a core, makes it possible to manufacture architected structures through the combination of multilayers of different materials and diameters. SMA cables combine the properties of conventional cables, such as relative tensile rigidity and flexibility in bending and torsion, with SMA features such as superelasticity. Furthermore, they have a reduced cost when compared to bulk bars of comparable size and a failure redundancy which is advantageous compared to bars and wires [8]. These features allow the use of SMA cables as self-sensing devices, self-centering devices and damping devices, and as sensors for health monitoring the integrity of structures and reinforced concrete applications.

Several investigations have been devoted to the study of the experimental response of SMA-based cables with different layouts, diameters and test conditions. The state-of-the-art survey that follows shows the main findings regarding SMA-cables in the last decade. Reedlunn et al. [8,9] showed in a detailed two-part study that different cable designs, which ultimately differ in helix angle, each have their own particular niche of mechanical and functional properties regarding isothermal, low strain-rate and superelastic applications. They showed that while a 7 × 7 NiTi cable is more suitable for high force applications, a 1 × 27 NiTi cable performs better for larger recoverable strain. Based on these results, Biggs and Shaw [10] explored the actuation performance (applying a dead load and varying temperature) of both cable designs, again observing significant differences between their mechanical behavior, and demonstrating that the 7 × 7 NiTi cable successfully scaled up the functional properties of a single NiTi wire. The strain-rate sensitivity, well-known for NiTi SMA, was studied by Mas et al. [11] for the 7 × 7 NiTi cable design, which showed significant temperature increase at strain rates from 10^−4^ s^−1^ to 10^−2^ s^−1^, resulting in increasing strain hardening at higher strain rates. Sherif and Ozbulut [12] investigated the functional fatigue life of NiTi SMA cables. They report on the very satisfactory cycling performance of the NiTi cables, especially compared to similar cross-section NiTi bars. Their tested cables showed only 0.76% of residual strain after cyclic loading at 10%. Fang et al. [13] explored the effect of heat treatments and loading protocols in the equivalent viscous damping (dissipated energy) of the cable under difference loading protocols at relatively low strain rates. Shi et al. [14] also studied the functional fatigue of 7 × 7 NiTi cables at different temperatures and strain amplitudes, observing that the equivalent viscous damping depends on both these parameters. Nevertheless, none of these studies evaluate the influence of considerably high loading frequencies (i.e., strain rates) on the functional properties of the NiTi cables, nor their influence on fatigue life.

In this context, the present study proposes to contribute to the SMA cable literature by investigating the influence of heat accumulation during high loading frequency cycling on the mechanical response of NiTi SMA micro cables. Based on the superelastic response in a dynamic regime, this work uses an experimental methodology for determination of the self-heating critical frequency and its influence on the thermomechanical behavior, functional properties and fatigue life of NiTi SMA micro cables. The micro cable was characterized through cyclic tensile tests at loading frequencies from 0.25 Hz to 10 Hz. The self-heating frequency was identified using a previously presented experimental methodology applied to the micro cable [7]. The structural and functional fatigues of the micro cable were evaluated as a function of the loading frequency for the failure of each individual wire.

## 2. Methodology

### 2.1. Experimental Procedure

#### 2.1.1. Cable Material and Geometry

The experiments were performed in the straight regions of smart-twist NiTi micro cables commercially supplied by the Beijing Smart Technology Co., Ltd., Beijing, China, as orthodontic arches with nominal composition of 55% Ni (wt.%). Figure 1 shows both an SEM image of the side view and a schematic drawing of a cross-section of the 1 × 7 micro cable consisting of six × 0.1354 mm wires wound in a left-handed helix around the straight core wire, and having an outer diameter of 0.4064 mm. The helix angle of −15° was estimated by SEM (model VEGA3 from TESCAN ANALYTICS, Fuveau, France) with BSE detector.

#### 2.1.2. Chemical Characterization

The chemical composition of the NiTi micro cable was analyzed using Energy Dispersive Spectrometer (EDS) (model X-act from Oxford Instruments, Abingdon, UK). The region analyzed was the external surface of a wire with 1000× magnification. The sample was cleaned with isopropyl alcohol in an ultrasonic bath for 10 min and dried with a heat gun.

#### 2.1.3. Thermal Characterization

The transformation temperatures of the NiTi micro cable were determined by Differential Scanning Calorimetry (DSC) (model Q20 from TA Instruments, New Castle, DE, USA). The temperature range was from −60 °C to 100 °C with a cooling/heating rate of 10 °C/min under a nitrogen atmosphere (ASTM F2004-17). The starting and final phase transformation temperatures were obtained by the tangent intersection method applied to the DSC peaks.

#### 2.1.4. Mechanical Characterization

To investigate the mechanical behavior of the material, a quasi-static tensile test was performed on the core wire. A sample with 12.6 mm of gauge length was submitted to isothermal load-unloading cycles at 7.5% maximum strain with a strain rate of 0.5%/min at temperatures ranging from 35 °C to 85 °C. The tensile tests were carried out in a Dynamic Mechanical Analyzer (DMA) (model Q800 from TA Instruments).

A mechanical pre-characterization in a quasi-static regime was performed on the NiTi micro cables to determine the pre-strain levels and oscillation amplitude. For this, tensile tests were performed according to ASTM F2516-18 standard. Therefore, a straight section of the sample was tensioned up to 6% in the first loading cycle and unloaded at 7 MPa residual stress with a strain rate of 0.04 min^−1^, while in the second cycle the sample was loaded until rupture with a strain rate of 0.4 min^−1^.

To investigate the behavior of the thermomechanical coupling of NiTi micro cables, the straight samples were submitted to a pre-cycling with 128 tensile cycles at 0.25 Hz of loading frequency in order to minimize the effect of the evolution of the superelastic behavior. The reason for this number of cycles is that we follow a 2n rule to accommodate the fact that the SMA mechanical behavior evolves much faster in the first cycles before reaching stabilization. After pre-cycling, the sample was submitted to 128 tensile sinusoidal cycles between 0.5 Hz and 10 Hz of loading frequency to evaluate the influence of frequency on the superelastic behavior. Both analyses were performed using the same amplitude range of 5% peak-to-peak, between 2.5% and 7.5%, as determined in the previous quasi-static test. Data were recorded with an acquisition rate of 1000 Hz.

The temperature variation during the tensile cyclic tests was monitored using a K-type micro thermocouple (100 μm in diameter) fixed with glue to the outer surface of the NiTi micro cable. Temperature evolution was recorded by a data acquisition system set with an acquisition rate of 100 Hz (model Quantum X from HBM, Darmstadt, Germany).

To investigate the mechanical behavior evolution regarding functional and structural fatigue of the NiTi micro cables, the samples were submitted to tensile sinusoidal cycles between 1 Hz and 10 Hz of loading frequency, at a strain amplitude of 5% peak-to-peak between 2.5% and 7.5%. Data were recorded with an acquisition rate of 500 Hz. 

For comparison purposes, a NiTi SMA wire with a diameter of 0.35 mm and a cross-sectional area equivalent to the cross-section area of the NiTi micro cable, and named reference element, was submitted to the same mechanical tests.

All samples had 30 mm gauge length and were tested at room temperature of 22 ± 2 °C in an electrical testing machine (model ElectroPuls E10000 from INSTRON, Norwood, MA, USA) equipped with a 10 kN load cell using strain control.

### 2.2. Theoretical Procedure

By using the results of the dynamic response of the NiTi SMA, the functional properties of dissipated energy per cycle ($E_{D}$, in MJ/m^3^) and equivalent viscous damping factor per cycle (ξ, in %) were determined, respectively, as follows:(1)$$E_{D}=\oint \sigma \;d\varepsilon$$
(2)$$\xi =\frac{E_{D}}{4\pi E_{SO}}\times 100$$
where σ represent stress and $E_{SO}$ is the dissipated energy of an equivalent linear element [15].

## 3. Results and Discussion

### 3.1. Experimental Analysis

#### 3.1.1. Chemical Composition

According to the EDS result shown in Figure 2a, the NiTi micro cable showed good surface homogeneity of the elements Ni and Ti. From the spectrum shown in Figure 2b, the NiTi SMA micro cable showed a composition of 53.8% Ni (wt.%).

#### 3.1.2. Transformation Temperatures

Figure 3a,b show the thermograms of the NiTi micro cable and the reference element, both in the as-received state, presenting final temperatures of austenite phase transformation around 17.3 °C and 18.5 °C, respectively. Therefore, under test conditions at room temperature, the NiTi SMA micro cable exhibits superelastic behavior. Both samples also exhibited R-phase as an intermediate phase in the forward-phase transformation (austenite to martensite), indicated by the low enthalpy of forward transformation (4.7 J/g and 6.1 J/g) and of reverse transformation (5.6 J/g and 6.7 J/g) for the NiTi micro cable and the reference element, respectively. Also, the thermal hysteresis was small for both the NiTi micro cable and the reference element (6.8 °C and 6.1 °C, respectively) [16,17].

#### 3.1.3. Isothermal Quasi-Static Superelastic Response of the Core Wire

Figure 4a shows the quasi-static superelastic stress vs. strain response of the NiTi SMA core wire under isothermal temperature conditions.

Overall, a reduction in the mechanical hysteresis and an increase in the residual strain are observed with increasing temperature. The increase in stresses of forward and reverse phase transformation follows the Clausius–Clapeyron relationship and can be estimated through the linear coefficient of the stress vs. temperature diagram, as shown in Figure 4b. For the start stresses for the forward and reverse phase transformation, the *C_M_* and *C_A_* coefficients are 5.54 MPa/°C and 6.28 MPa/°C, respectively.

It is also possible to observe that the plateau stresses are well defined during the forward and reverse phase transformations. This phenomenon is associated with Lüders-type deformation behavior due to the strain heterogeneity along the sample during phase transformation [18].

#### 3.1.4. Quasi-Static Superelastic Response

Figure 5a,b show the quasi-static superelastic stress vs. strain response at room temperature of the NiTi SMA micro cable and the reference element, respectively. It is worth noting that we considered the length between grips equal to the gauge length of 30 mm due to sample limitations, although the ASTM F2516 standard recommends a minimum distance between the grips shall of 150 mm when using the motion of the crosshead to calculate the strain in samples with diameter less than or equal to 0.2 mm. Upper Plateau Strength (UPS) was determined as 398.09 MPa and 476.32 MPa at a strain of 3.0% during the initial loading in the samples of the NiTi SMA micro cable and the reference element, respectively, while the Lower Plateau Strength (LPS) was determined as 114.75 MPa and 178.46 MPa at a strain of 2.5% during the unloading in the samples of the NiTi SMA micro cable and the reference element, respectively.

The NiTi SMA micro cable exhibited a small inclination in the first cycle of the loading (0.44% of strain and 167.6 MPa of stress), referent to the R-phase as an intermediate stress-induced phase in the forward-phase transformation [19].

From the results shown in Figure 5, a strain range ranging from *ε_min_* = 2.5% to *ε_max_ =* 7.5% was selected as the oscillation amplitude and a strain of 5% was selected as the pre-strain for the cyclic tension tests in the dynamic regime.

The definition of parameters such as pre-strain and amplitude of oscillation is a fundamental step for the designer during the implementation of structural elements for applications in dynamic regimes. If they are poorly defined, it can lead to structural element inefficiency and even premature failure.

#### 3.1.5. Superelastic Response in Dynamic Regime

Figure 6 shows the superelastic loops of the 1st and 128th cycles of the NiTi SMA micro cable for frequencies from 0.25 Hz to 10 Hz. As observed, for frequencies 0.25 Hz and 0.5 Hz, the peak stress for the 128th cycle is lower than for 1st cycle. The peak stresses for the 128th and 1st cycles are 442.84 MPa and 467.04 MPa, respectively, for 0.25 Hz, which is a reduction of approximately 5.5%. At 0.5 Hz, the peak stresses for the 128th and 1st cycles are 443.35 MPa and 462.25 MPa, respectively, a reduction of approximately 4.3%. Indeed, the peak stress decrease is due to the redistribution of internal stresses during stress-induced phase transformation, mainly caused by local accumulation of dislocation slip [19]. It is worth noting, however, that the literature on the subject shows sufficient SMA mechanical stabilization around 50 cycles [11], and even sooner if the minimal stress variation is tolerated [10,13], since the main portion of the stabilization occurs during the first cycles.

From the frequency of 1 Hz, the peak stress for the 128th cycle is higher than for the 1st cycle and evolves with the increase of the loading frequency for the other tests. At 1 Hz, the peak stresses for the 128th and 1st cycles are 452.01 MPa and 451.67 MPa, respectively, an increase of approximately 0.07%. At 9 Hz, the NiTi SMA micro cable reaches the highest peak stress value, and at 10 Hz it drops. This reduction may be related to functional fatigue in the NiTi SMA micro cable properties. 

This frequency-dependent behavior is linked to the accumulation of latent heat generated during successive cycles of stress-induced forward phase transformation, leading to an increase in the internal temperature of the NiTi SMA micro cable and, consequently, a reduction in dissipated energy and an upward displacement of the superelastic loops due to the Clausius–Clapeyron relation.

Figure 7 shows the superelastic loops of the 1st and 128th cycles of the reference element for frequencies from 0.25 Hz to 10 Hz. As observed, for frequencies 0.25 Hz, 0.5 Hz and 1 Hz, the peak stress for the 128th cycle is lower than for the 1st cycle. The peak stresses for the 128th and 1st cycles are 458.93 MPa and 498.41 MPa, respectively, for the frequency of 0.25 Hz, a reduction of approximately 8.6%. At 0.5 Hz, the peak stresses for the 128th and 1st cycles are 520.18 MPa and 504.17 MPa, respectively, a reduction of approximately 3.2%. For 1 Hz, the peak stresses for the 128th and 1st cycles are 546.65 MPa and 538.87 MPa, a reduction of approximately 1.4%.

From 2 Hz, the peak stress for the 128th cycle is higher than for the 1st cycle and evolves with the increase of the loading frequency. At 2 Hz, the peak stresses for the 128th and 1st cycles are 550.28 MPa and 547.74 MPa, respectively, an increase of approximately 0.46%. At 10 Hz, the reference element reaches the highest peak stress value. This behavior is linked to the accumulation of latent heat generated during successive cycles of forward phase transformation, leading to an increase in the internal temperature of the reference element and, consequently, a reduction in dissipated energy and an upward displacement of the superelastic loops due to the Clausius–Clapeyron relation.

As discussed, the internal area of the superelastic loop (dissipated energy) of the reference element is visibly smaller than that of the cable. This can be understood as an additional portion of energy dissipation being originated during the friction between the wires in the NiTi SMA micro cable.

From the superelastic stress vs. strain cycles, the percentage peak stress difference (Δ*σ*) between the 128th and 1st cycles is calculated and shown in Figure 8. The blue dots indicate the frequencies at which Δ*σ* was negative (below the dashed line), while the red dots indicate the frequencies at which Δ*σ* was positive (above the dashed line).

For the micro cable (Figure 8a), the Δ*σ* value is negative, i.e., the peak stress for the 128th cycle is lower than the peak stress for the 1st cycle, at the frequencies of 0.25 Hz and 0.5 Hz. From 1 Hz, Δ*σ* is positive and increases almost linearly with the loading frequency up to 9 Hz, when the stress increases approximately 25.5%. 

Similarly, from the superelastic stress vs. strain cycles shown in Figure 7, the Δ*σ* value is negative for frequencies of 0.25 Hz to 1 Hz for the reference element, as shown in Figure 8b. From 2 Hz, Δ*σ* is positive and increases almost linearly with the loading frequency, reaching approximately a 15% increase in the reference element.

The intersection between Δ*σ* = 0 line and the Δ*σ* curve for various cycling frequencies identifies the self-heating frequency (*f_c_*) above which the material accumulates latent heat, increasing its internal temperature and consequently changing its functional properties. The methodology for determining the parameter *f_c_* used here is based on the work of de Souza et al. [7] in which they analyzed a NiTi SMA Belleville spring under cyclic compression tests under various loading frequencies.

The *f_c_* values for the NiTi SMA micro cable and the reference element analyzed in this work are 0.99 Hz and 1.76 Hz, respectively. Note that the *f_c_* value is smaller for the NiTi SMA micro cable than for the reference element, showing a higher sensitivity to the increase in frequency. This is probably due to the friction between the wires of the micro cable, which provokes an increase in the internal temperature and does not occur in the reference element.

The *f_c_* value depends on several factors, such as specific convection and conduction heat transfer conditions, the amplitude of oscillation, pre-strain and the geometric parameters of the NiTi SMA micro cable. It is up to the designer to analyze the environmental conditions of a specific application and take them into account to reach the energy dissipation requirements. 

Figure 9 illustrates the self-heating phenomenon in the NiTi SMA micro cable by showing the temperature signal throughout cycling. The upward displacement of the superelastic loops observed in Figure 6 is related to self-heating, where SMA accumulates the latent heat generated due to the stress-induced forward phase transformation that is not completely dissipated during the stress-induced reverse phase transformation.

Note that the mean temperature increases with the evolution of the cycles, and the temperature variation amplitude reduces with the increase of the loading frequency. The temperature signal starts above the room temperature of 22 ± 2 °C and then decreases. This is due to the 5% pre-strain applied at a low strain rate (equal for all tests), which increases the starting temperature, and soon after the unloading to 2.5% and then load to 7.5%, closing the first cycle.

For the 0.25 Hz frequency, the peak temperature for the 128th cycle is 27.38 °C, while for the 10 Hz frequency it is 49.25 °C, increasing approximately 80%. Note that the 128 cycles were not enough to reach thermal equilibrium. Still, it may represent real application situations in which the phenomena are of short duration, lasting for a few tens of seconds. 

Similarly, Figure 10 illustrates the self-heating phenomenon in the reference element by showing the temperature signal throughout cycles. The upward displacement of the superelastic loops observed in Figure 7 is related to self-heating, where SMA accumulates the latent heat generated due to the stress-induced forward phase transformation that is not completely dissipated during the stress-induced reverse phase transformation.

Note that the temperature increases with the evolution of the cycles, and a reduction is observed for the amplitude with the increase of the loading frequency. The temperature signal starts above the room temperature of 22 ± 2 °C and then decreases. This is due to the 5% pre-strain applied at a low strain rate (equal for all tests), which increases the starting temperature, and soon after the unloading to 2.5% and then load to 7.5%, closing the first cycle.

At 0.25 Hz, the peak temperature for the 128th cycle is 28.48 °C, while for the 10 Hz frequency it is 43.2 °C, increasing approximately 52%. Note that the 128 cycles were not enough to reach thermal equilibrium.

We observed an exponential decreasing relationship between the temperature amplitude (*T_amp_*) for the 128th cycle of the NiTi SMA micro cable and the loading frequency (*f*), as shown in Figure 11a. An abrupt decrease is observed up to 5 Hz, from 8.65 °C at 0.25 Hz to 1.01 °C at 5 Hz. An asymptotic regression model was used, with coefficients a = 0.649, b = −8.798 and c = 0.538, being able to estimate with good precision (R^2^ = 0.98) the *T_amp_* from a certain *f*. Figure 11b shows the linear relationship between the mean temperature (*T_mean_*) for the 128th cycle and the frequency (*f*) from 1 Hz. It is observed that *T_mean_* increases almost linearly with *f*, from 25.88 °C at 1 Hz to 48.98 °C at 10 Hz, with coefficients a = 25.54 and b = 2.06 °C/Hz (R^2^ = 0.84). It is worth noting that these fitting relations are valid for the entire system (micro cable, grips, atmosphere) providing an order of magnitude for the relationships between temperature variables *T_amp_* and *T_mean_* with *f*.

### 3.2. Theoretical Analysis

#### 3.2.1. Functional Properties

As mentioned before, from the superelastic dynamic response of the NiTi SMA micro cable shown in Figure 6, the functional properties such as dissipated energy per cycle and equivalent viscous damping factor per cycle were obtained. 

Figure 12a shows the behavior of *E_D_* vs. cycle number for the analyzed frequencies. The cycles are plotted at each 2n cycle (*n* = 0, 1, 2, 4, …, *n*) to facilitate visualization. Overall, *E_D_* decreases over the cycles and with an increase in the loading frequency. For the frequency of 0.25 Hz, the reduction of *E_D_* over the cycles is more accentuated, reaching approximately 28% between the 1st and 128th cycles (from 9.98 MJ/m^3^ to 7.16 MJ/m^3^). At 0.5 Hz, the reduction was approximately 17% (from 7.59 MJ/m^3^ to 6.29 MJ/m^3^). From 3 Hz, the percentage difference between the 1st and 128th cycles remains almost constant with a value of approximately 20%. As for the variation with the frequency, the *E_D_* variation between 1 Hz and 10 Hz at the 128th cycle reached approximately 45% (5.87 MJ/m^3^ and 3.22 MJ/m^3^, respectively).

Figure 12b shows the variation of ξ with cycle number for each analyzed frequency. The parameter ξ is directly related to *E_D_* and, therefore, follows the same behavior. The highest values of the parameter ξ are observed at 0.25 Hz, which are 6% and 4.96% for the 1st and 128th cycles, respectively, while the lowest values are observed at 10 Hz, which are 2.91% and 1.49% for the 1st and 128th cycles, respectively. According to Speicher et al. [15], values of ξ around 5–10% are typical in structural engineering. Although the geometry, material and test conditions are different for the NiTi SMA cable studied by Fang et al. [13] and Shi et al. [14], ξ values are in the same range.

Similarly, from the superelastic dynamic response of the reference element shown in Figure 7, the functional properties such as dissipated energy per cycle and equivalent viscous damping factor per cycle were obtained. Figure 13a shows the behavior of *E_D_* vs. cycle number for each analyzed frequency. The cycles are plotted at each 2n cycle (*n* = 0, 1, 2, 4, …, *n*) to facilitate visualization. Overall, *E_D_* decreases with cycling and with an increase in the loading frequency. At 0.25 Hz, the reduction of *E_D_* with cycling is more accentuated, reaching approximately 28% between the 1st and 128th cycles (from 8.78 MJ/m^3^ to 6.28 MJ/m^3^). At 0.5 Hz, the reduction was approximately 19% (from 6.44 MJ/m^3^ to 5.2 MJ/m^3^). The highest reduction was observed at 10 Hz, reaching approximately 30% (from 3.03 MJ/m^3^ to 2.12 MJ/m^3^). As for the variation with the loading frequency, the *E_D_* variation between 1 Hz and 10 Hz at the 128th cycle reached approximately 50% (4.22 MJ/m^3^ and 2.12 MJ/m^3^, respectively).

Figure 13b shows the variation of ξ with cycle number for each analyzed frequency. The highest values are observed for the frequency of 0.25 Hz, which are 4.67% and 3.89% for the 1st and 128th cycles, respectively, while the lowest values are observed at 10 Hz, which are 1.60% and 0.84% for the 1st and 128th cycles, respectively.

In Figure 14 we compare the percentage difference (Δ, in %) of functional properties such as *E_D_* and ξ between the NiTi SMA micro cable and the reference element. Figure 14a shows the variation of Δ*E_D_* vs. cycle number for the analyzed frequencies. Overall, Δ*E_D_* remains practically constant during cycling in the range between 15% and 30%. However, it tends to increase with increasing loading frequency. The lowest value observed was at 0.25 Hz with an average of 12.5% during cycling. The highest value (32.4%) occurred at 7 Hz. Analogously, Figure 14b shows the variation of Δξ. The Δξ values remain practically constant with cycling in the range between 20% and 45% until the 16th cycle. Afterwards, Δξ behavior was different for different frequencies, reaching a minimum of 23% at 0.25 Hz and a maximum of 47% at 7 Hz.

Although Δ*E_D_* and Δξ varied significantly, these results confirm that the linear elements with multifilament geometry are more advantageous compared to bulk elements when it comes to energy dissipation due to the extra amount of friction between the filaments. It is worth noting that other factors can influence the micro cable superiority regarding the observed functional properties, such as the higher mechanical hysteresis in relation to that observed for the reference element (see Figure 5).

#### 3.2.2. Structural and Functional Fatigue

Fatigue life was evaluated for the NiTi SMA micro cable at loading frequencies above *f_c_*. Figure 15 and Figure 16 show the superelastic stress vs. strain response from the first to the seventh individual failures (N*_if_*, with *i* = 1 to 7) for loading frequencies from 1 Hz to 10 Hz. Each superelastic curve identified in the 3D graph corresponds to the cycle prior to failure. The initial cross-section area was used for the stress calculation.

The peak stress until the first failure increases with the increase of the loading frequency up to 8 Hz, from 444.89 MPa at 1 Hz to 622.50 MPa at 8 Hz, as shown in the inserted graphs. For the 9 Hz and 10 Hz frequencies (563.27 MPa and 528.01 MPa, respectively), the peak stresses up to the first failure are lower, probably due to a marked degradation of the micro cable’s functionality at these higher loading frequencies.

It is also possible to observe an almost linear reduction of the peak stress with the N*_if_*. Overall, the NiTi SMA micro cable is able to maintain high dissipating energy levels until the third failure, especially at higher frequencies. This demonstrates that the NiTi SMA micro cable applied as a structural element or as a sensor in monitoring the integrity of structures has an additional safety factor, not requiring immediate replacement after the failure of the first filaments, unlike a bulk element such as a wire or a spring, for example.

Fatigue life was evaluated for the reference element at loading frequencies above *f_c_*. Figure 17 and Figure 18 show the superelastic stress vs. strain response over failure cycles. The cycles are plotted at each 2n cycle (*n* = 0, 1, 2, 4, …, *n*) to facilitate visualization for loading frequencies from 1 Hz to 10 Hz.

As observed, for frequencies 1 Hz and 2 Hz the peak stresses for the last cycle (the 2097th and the 4194th, respectively) are lower than for the first cycle. The peak stresses for the 2097th and for the first cycles are 495.8 MPa and 532.27 MPa, respectively, for the frequency of 1 Hz, a reduction of approximately 6.8%. At 2 Hz, the peak stresses for the 4194th and for the 1st cycles are 511.64 MPa and 545.1 MPa, respectively, a reduction of approximately 6.1%.

From 3 Hz to 6 Hz, the peak stresses values referring to the 2n cycles increase throughout cycling, with a negligible difference between the first and last cycles. However, from 7 Hz the last cycle remains above the first cycle, indicating a severe condition of mechanical loading, which leads to a rapid degradation of the functional properties of the reference element.

The functional fatigue of the NiTi SMA micro cable was also evaluated, based on the superelastic loop evolution shown in Figure 15 and Figure 16. From this, Figure 19 shows the behavior of *E_D_* vs. N*_if_* for the analyzed frequencies. Overall, *E_D_* decreases with each failure and with increasing loading frequency. For the first failure, the reduction of *E_D_* between 1 Hz and 10 Hz reached approximately 41% (from 5.05 MJ/m^3^ to 2.97 MJ/m^3^). For the seventh failure, the reduction reached approximately 29% (from 3.64 MJ/m^3^ to 2.56 MJ/m^3^). For some loading frequencies, the NiTi SMA micro cable dissipates up to 1 MJ/m^3^ of mechanical energy immediately prior to the total rupture (seventh failure).

The structural fatigue of the NiTi SMA micro cable was evaluated based on the number of cycles up to the failure of the first filament (first failure, N*_f_* × 10^3^) from 1 Hz to 10 Hz. We highlight that the presented fatigue study, especially concerning structural fatigue, is preliminary, since no statistical analysis was performed. Nevertheless, the results bring an order of magnitude to its dependence on loading frequency, and a relevant comparison with the wire used as reference element.

In Figure 20a, as expected, a reduction in the *N_f_* is observed with an increase in the loading frequency. At 1 Hz, the NiTi SMA micro cable supported 3602 cycles before failure. An abrupt reduction in the number of *N_f_* from 2 Hz (2041 cycles) was observed, probably due to an increase in the internal temperature of the NiTi SMA micro cable, since at 1 Hz (slightly higher than *f_c_*), the micro cable reached a *T_mean_* = 25.88 °C for 128 cycles (see Figure 11b). At 2 Hz, the *T_mean_* was 30.53 °C and the micro cable reached 2041 cycles. At 8 Hz, the *T_mean_* was 44.49 °C and the micro cable reached 1215 cycles. Note that this *T_mean_* does not represent the temperature at which the micro cable was cycled, since the 128 cycles were not enough to stabilize the temperature.

Similarly, from the superelastic stress vs. strain cycles shown in Figure 17 and Figure 18, the structural fatigue of the reference element was evaluated from the number of cycles to failure (N*_f_* × 10^3^), from 1 Hz to 10 Hz, as shown in Figure 20b. In general, however, a reduction in *N_f_* is observed with an increase in the loading frequency, with values between 3 × 10^3^ and 6 × 10^3^ cycles until failure, while the NiTi SMA micro cable reached values of up to 2 × 10^3^ cycles for the first failure from 2 Hz to 10 Hz.

The NiTi SMA micro cable and the reference element reached a low cycle fatigue life, which is of the order of 10^3^ cycles [1]. For reference, Sherif and Ozbulut [12,20] observed the first failures around the 3 × 10^3^ cycles in a superelastic multi-layered NiTi cable under tension mode at 7% strain amplitude.

From 2 Hz, a linear regression model was used to estimate the relationship between the frequency (*f*) and the failure cycles (N*_f_* × 10^3^) for the first failure of the NiTi SMA micro cable. It is observed that *f* decreases almost linearly with N*_f_*, with an angular coefficient equal to 26.47 Hz/N*_f_* (R^2^ = 0.52). Similarly, a linear regression model was used to estimate the relationship between *f* and N*_f_* for the reference element, with an angular coefficient of 13.04 Hz/N*_f_* (R^2^ = 0.51). Even with the low precision of the resulting R-Square, it is possible to observe that the NiTi SMA micro cable is more sensitive to the loading frequency than the reference element. Our hypothesis for the reduced fatigue life of the NiTi SMA micro cable compared to the reference element is due to the friction between the strands, which accelerates the crack initiation.

Several factors affect the fatigue life of Ni-Ti SMA, such as oscillation amplitude, loading frequency, loading mode [17,21,22] and surface finish, for example. Due to these and other factors, the evaluation of cable fatigue life is complex because of the multifilament geometry of twisted wires wrapped around a core wire, which causes friction, as previously pointed out. In addition, the manufacturing process can induce defects that reduce the fatigue life of these cables, as shown, for example, in Figure 21. Similar defects were noted by Reedlunn et al. [9].

## 4. Conclusions

The thermodynamic coupling of NiTi SMA micro cables was evaluated at low and high loading frequencies, from 0.25 Hz to 10 Hz. The temperature and mechanical behavior of the micro cable were monitored during cycling. Results were compared with those of a single wire with similar cross-section area to analyze the influence of the extra portion of friction energy dissipation present in the micro cable. The conclusions are summarized as follows:As expected, with an increasing number of cycles, an increase in internal temperature and a decrease in energy dissipation capacity of both micro cable and wire reference element is observed;The self-heating frequency, above which the SMA element experiences a significant accumulation of latent heat and can no longer sustain an isothermal cycling, was approximately 1 Hz for the micro cable, while it was 1.76 Hz for the reference wire element. The difference is mostly due to the extra heat release caused by friction in the micro cable, causing it to heat faster. Determining the self-heating frequency of SMA elements is of utmost importance when dynamic applications are in sight;The temperature reached after 128 cycles increased 80% (from 27 °C to 49 °C) from 0.25 Hz to 10 Hz for the micro cable, and 52% (28 °C to 43 °C) for the wire. The significantly stronger heating in the micro cable is associated with friction between the wires;Dissipated energy per cycle and equivalent viscous damping factor decrease much faster with cycling as the loading frequency increases. This decrease reached 49% at 10 Hz for only 128 cycles, while it was 17% at 0.25 Hz. Very little difference was observed between the micro cable and the wire;Due to the rapid increase in the temperature of the micro cable with increasing frequencies, which significantly stiffens the SMA due to the Clausius–Clapeyron law, the structural fatigue life (up to the failure of the first filament of the micro cable) was half the fatigue life of the wire. However, due to the cable geometry, the delayed failure compared to that of a single wire is much more beneficial for sensoring applications.

## Figures and Tables

**Figure 1 sensors-22-08045-f001:**
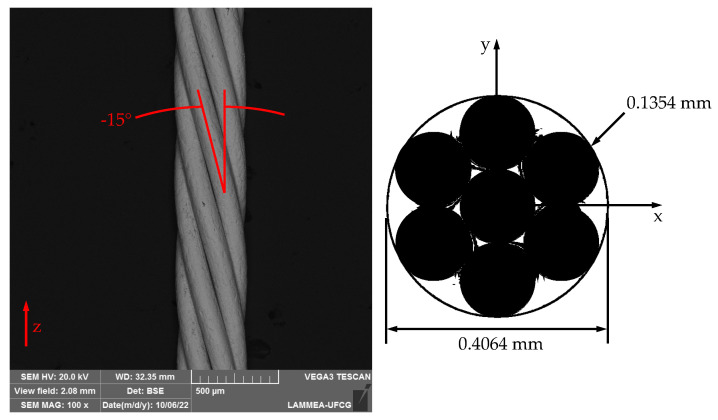
SEM image of the side view and schematic drawing of a cross-section of the NiTi micro cable.

**Figure 2 sensors-22-08045-f002:**
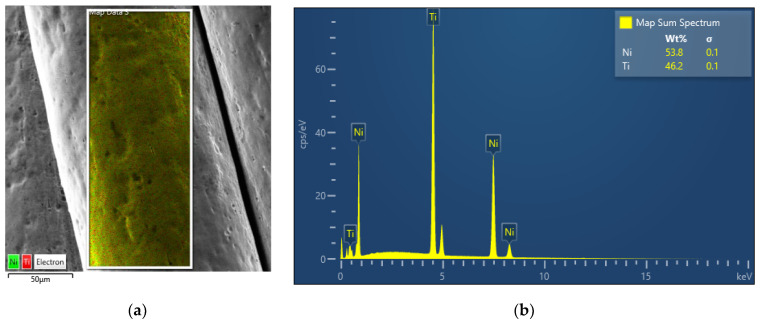
EDS elemental map analysis for the NiTi SMA micro cable. (**a**) EDS layered image. (**b**) Spectra with weight composition.

**Figure 3 sensors-22-08045-f003:**
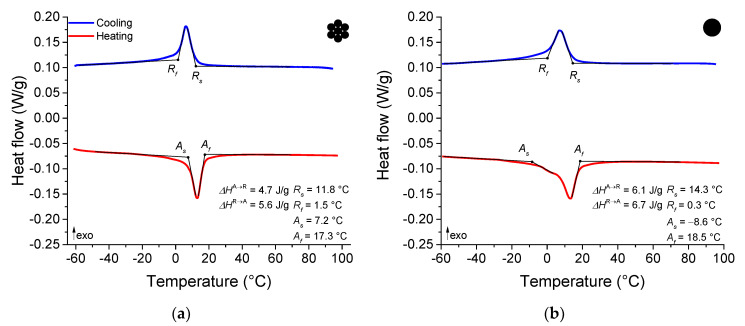
Thermal behavior characterized by DSC. (**a**) NiTi SMA micro cable. (**b**) Reference element.

**Figure 4 sensors-22-08045-f004:**
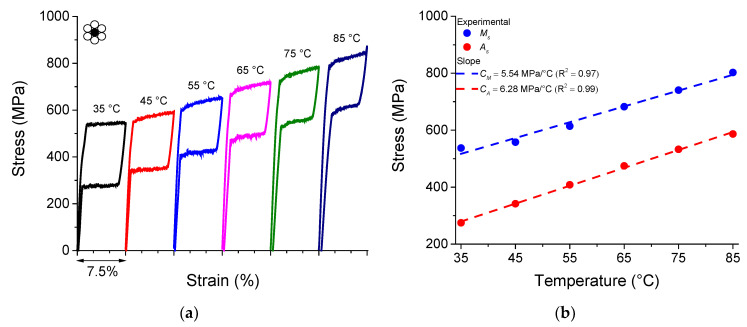
(**a**) Superelastic stress vs. strain response of the NiTi SMA core wire for different temperatures. (**b**) Phase diagram of NiTi SMA core wire.

**Figure 5 sensors-22-08045-f005:**
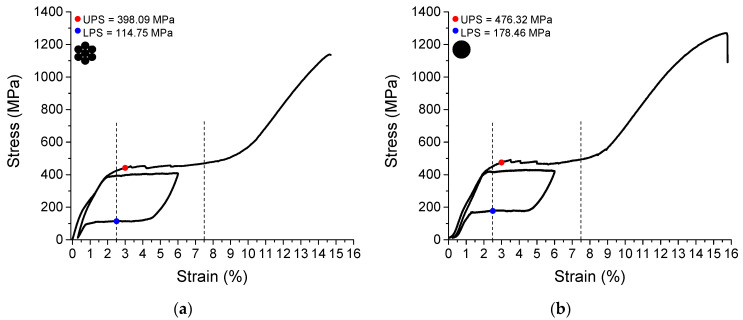
Quasi-static superelastic response. (**a**) NiTi SMA micro cable. (**b**) Reference element. The vertical lines indicate the oscillation amplitude selected for the cyclic tension tests.

**Figure 6 sensors-22-08045-f006:**
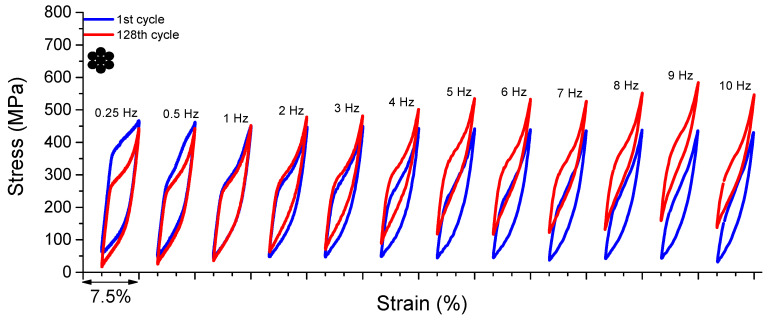
Dynamic response of the superelastic behavior of the NiTi SMA micro cable for loading frequencies from 0.25 Hz to 10 Hz.

**Figure 7 sensors-22-08045-f007:**
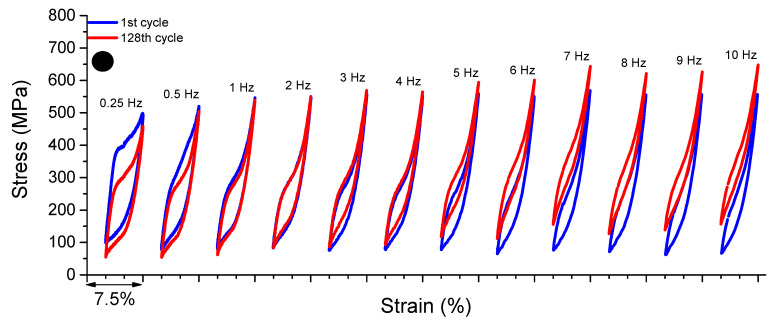
Dynamic response of the superelastic behavior of the reference element for loading frequencies from 0.25 Hz to 10 Hz.

**Figure 8 sensors-22-08045-f008:**
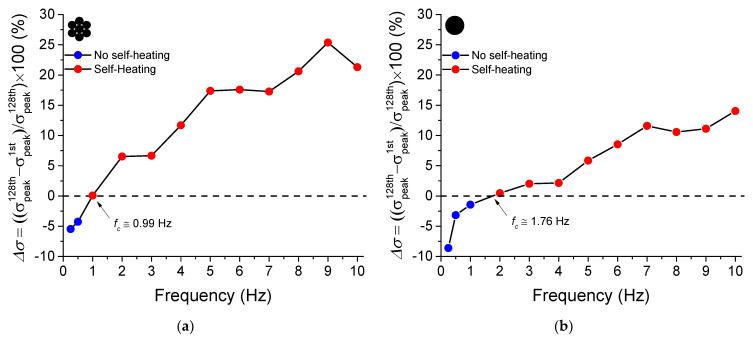
Critical frequency of self-heating. (**a**) SMA NiTi micro cable. (**b**) Reference element.

**Figure 9 sensors-22-08045-f009:**
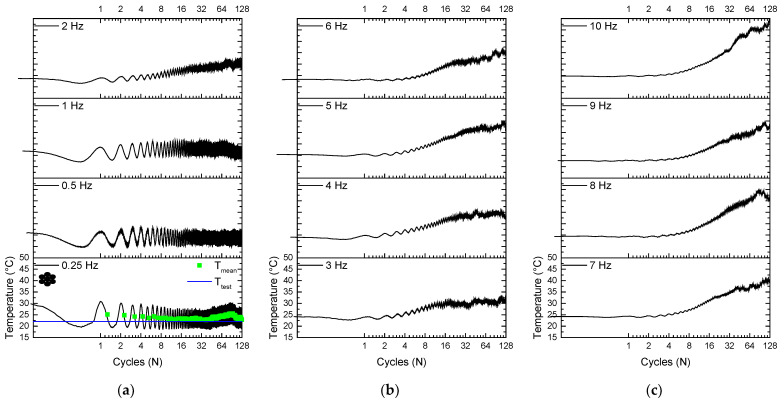
Temperature response during cycling of the NiTi SMA micro cable. (**a**) From 0.25 Hz to 2 Hz. (**b**) From 3 Hz to 6 Hz. (**c**) From 7 Hz to 10 Hz.

**Figure 10 sensors-22-08045-f010:**
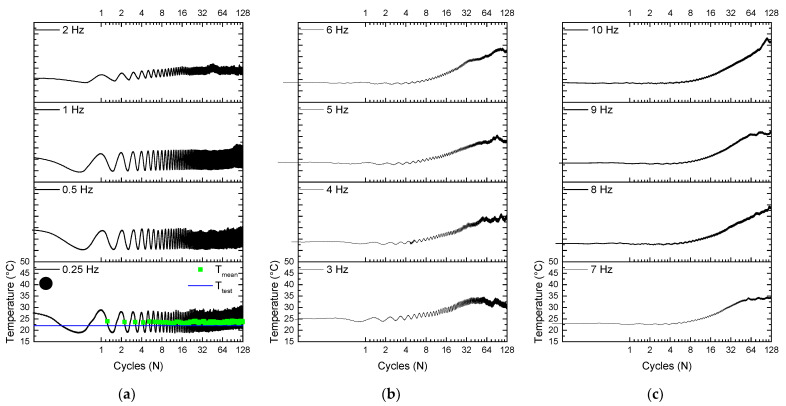
Temperature response during cycling of the reference element. (**a**) From 0.25 Hz to 2 Hz. (**b**) From 3 Hz to 6 Hz. (**c**) From 7 Hz to 10 Hz.

**Figure 11 sensors-22-08045-f011:**
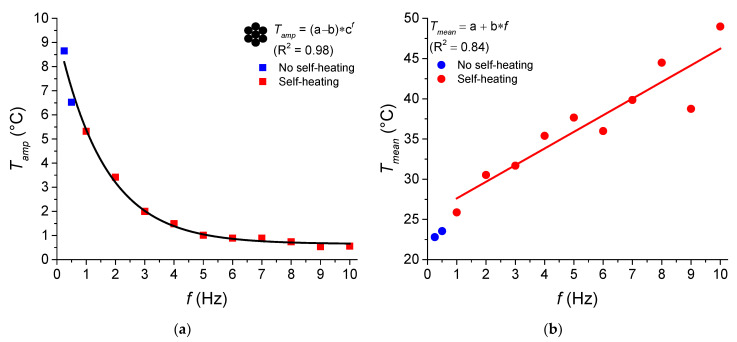
Temperature behavior for the 128th cycle of the NiTi SMA micro cable as a function of loading frequency (*f*). (**a**) Temperature amplitude (*T_amp_*). (**b**) Mean temperature (*T_mean_*).

**Figure 12 sensors-22-08045-f012:**
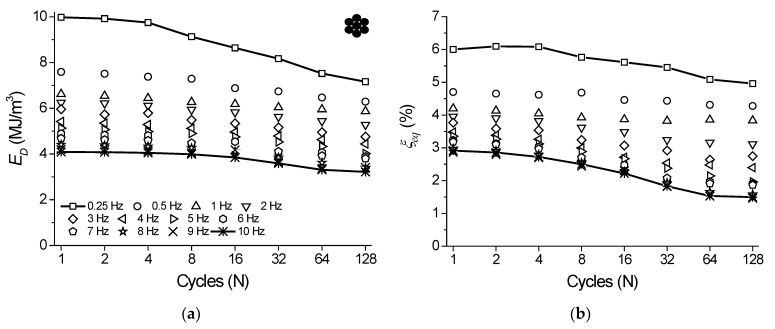
Functional properties variation with cycling as a function of the loading frequencies for the NiTi SMA micro cable. (**a**) Dissipated energy. (**b**) Equivalent viscous damping factor.

**Figure 13 sensors-22-08045-f013:**
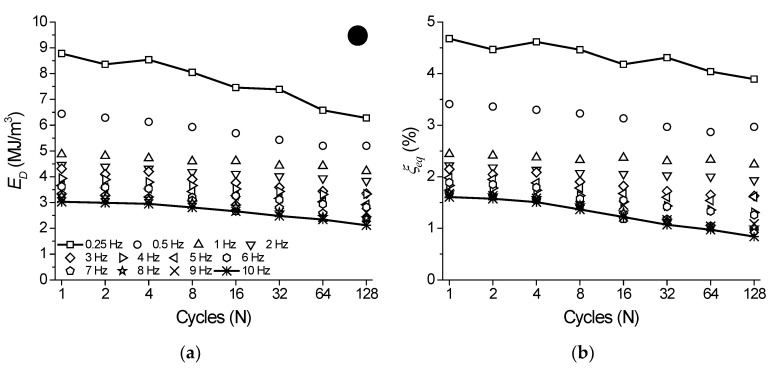
Functional properties variation with cycling as a function of the loading frequencies for the reference element. (**a**) Dissipated energy. (**b**) Equivalent viscous damping factor.

**Figure 14 sensors-22-08045-f014:**
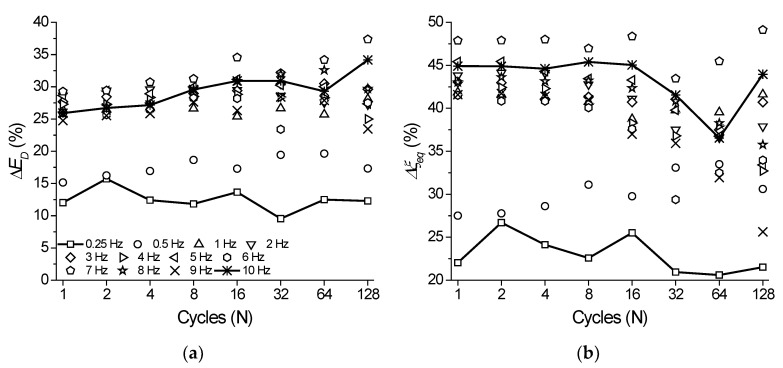
Percentage difference between the NiTi SMA micro cable and the reference element functional properties during cycling. (**a**) Dissipated energy. (**b**) Equivalent viscous damping factor.

**Figure 15 sensors-22-08045-f015:**
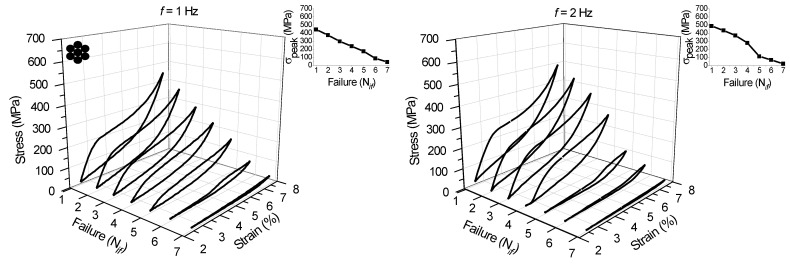
Stress vs. strain superelastic response immediately prior to the failure of the filaments of the SMA NiTi micro cable from 1 Hz to 6 Hz.

**Figure 16 sensors-22-08045-f016:**
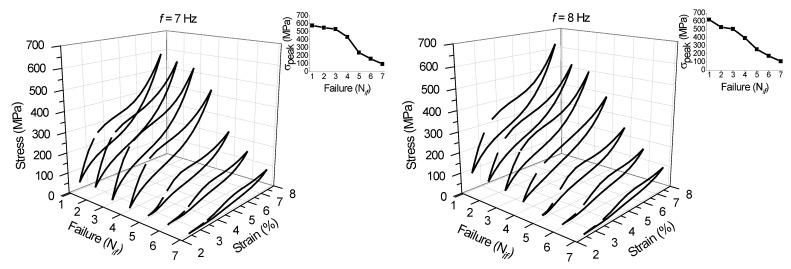
Stress vs. strain superelastic response immediately prior to the failure of the filaments of the SMA NiTi micro cable from 7 Hz to 10 Hz.

**Figure 17 sensors-22-08045-f017:**
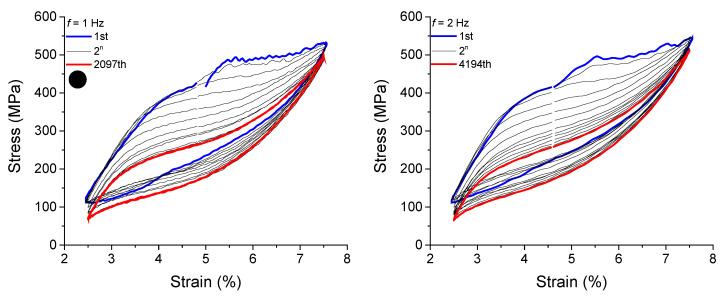
Stress vs. strain superelastic response of the reference element for loading frequencies from 1 Hz to 6 Hz.

**Figure 18 sensors-22-08045-f018:**
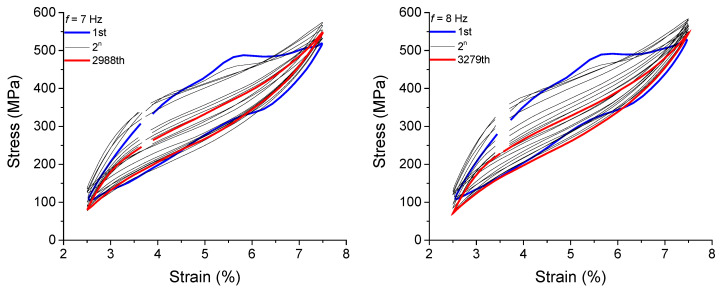
Stress vs. strain superelastic response of the reference element for loading frequencies from 7 Hz to 10 Hz.

**Figure 19 sensors-22-08045-f019:**
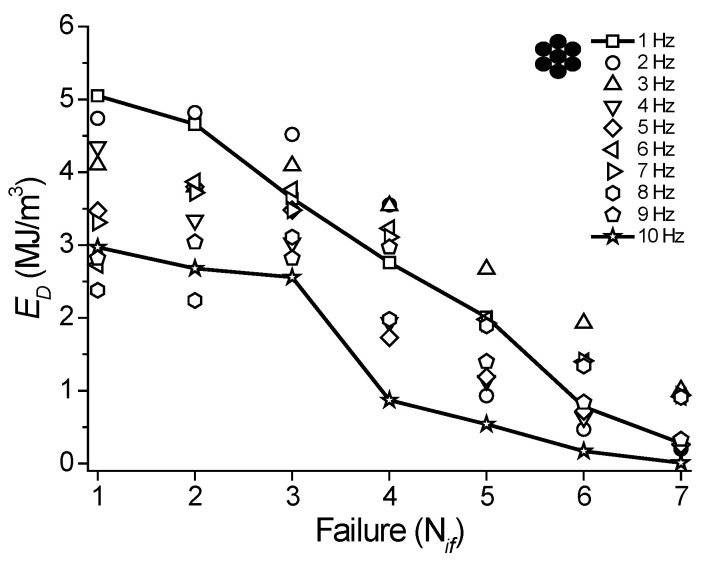
Dissipated energy evolution with the failure cycles for the loading frequencies from 1 Hz to 10 Hz for the NiTi SMA micro cable.

**Figure 20 sensors-22-08045-f020:**
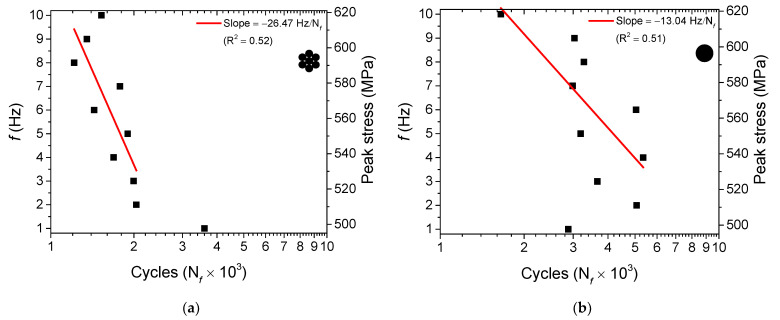
Fatigue life at different loading frequencies between *ε_min_* = 2.5% and *ε_max_* = 7.5%. (**a**) SMA NiTi micro cable. (**b**) Reference element.

**Figure 21 sensors-22-08045-f021:**
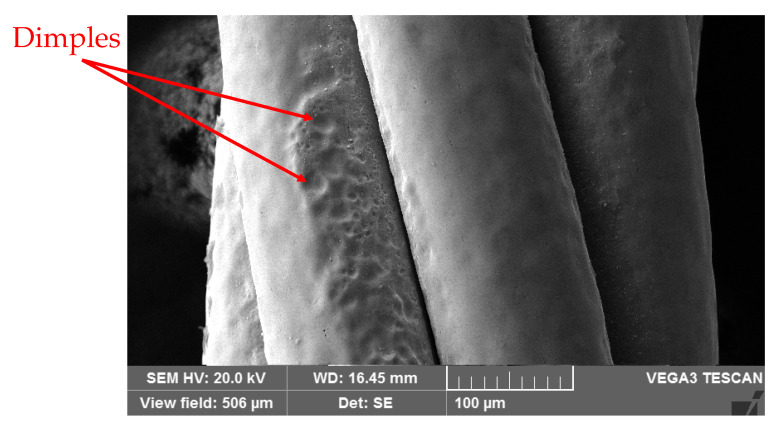
SEM image showing dimples on the surface of the NiTi SMA micro cable which contribute to a lower structural fatigue life.

## Data Availability

The data that support the findings of this study are available upon request from the authors.

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
