# Peer review of "NiTi SMA Superelastic Micro Cables: Thermomechanical Behavior and Fatigue Life under Dynamic Loadings"

_sensors, 2022, doi:10.3390/s22208045_

Round 1

Reviewer 1 Report

Research is very interesting. I suggest a few points than can improve the quality of the paper before the publication:

 - Make a revision of the text (PlagScan document attached – marked

 - The discussion should be supported with more references.

 - Please, make your statements concise in the Conclusion section.

 - In the light of presented facts, I think that the paper requires major revision to reach the quality standards of this Journal.

Author Response

Dear Reviewer.

Attached are the responses to the comments point by point and the article with the highlighted changes (red).

We thank you for your dedication to evaluating our work.

Sincerely,

Paulo C.S. da Silva

Dear Reviewer

We send one marked copy of the manuscript in which it may be seen that all the suggestions have been taken into account. The corrections are marked in the electronic version of the paper that was improved.

Reviewer#1

Research is very interesting. I suggest a few points than can improve the quality of the paper before the publication:

Point 1: Make a revision of the text (PlagScan document attached – marked.

Response 1: We thank you for the plagiarism analysis. It is our understanding that the relatively high level of plagiarism identified by the PlagScan software is due to an extensive internet scan, which was not limited to the references cited in the article or the literature in the area. This was verified after, for example, a match was found with an article on geology, which is a subject far from those within the paper. Besides, it appears to us that even the titles in the reference section, as well as the axis in the presented graphs, were considered matches by the software.

Point 2: The discussion should be supported with more references.

Response 2: We have revised the discussion section and we added some points to our discussions. Some references were added, however, as some were also removed (mainly from the Introduction section), the final number of references is the same.

Point 3: Please, make your statements concise in the Conclusion section.

Response 3: Indeed, we have completely remodeled our Conclusion section, now pointing out the main findings and observations in the form of a list, which is hopefully more objective. As the Introduction section was also partially remodeled, we also checked that both sections are coherent.

We trust that the manuscript will meet with your approval, but should any doubt remain, please let us know.

Thank you for your attention.

Best regards,

Paulo C.S. Silva, Estephanie N.D. Grassi, Carlos J. Araújo, João M.P.Q. Delgado and António G.B. Lima

Reviewer 2 Report

1.      The influence of temperature effect on the energy dissipation properties of SMA is well known. The reasons why Thermomechanical coupling evaluation should be further explained,

2.      The effect of attenuation of vibrations is usually evaluated mainly through the hysteresis curves of energy dissipation of force and displacement, not the stress and strain curves. The main purpose of this research should be to use SMA for health monitoring instead of attenuation of vibration. The purpose of the research in this article should be further clarified

3.      “However, in dynamic regime applications, such as vibration attenuation, where repetitive superelasticity cycles are required in loading frequencies sufficiently high, this thermal equilibrium is affected and the material experiences a gradual accumulation of heat, which consequently leads to an increase in the internal temperature. This thermomechanical coupling is the cause of the strain rate dependence observed in SMA”. The situation described in this sentence is not consistent with the requirements of the vibration attenuation.

4.      The research significance and conclusions of this paper are not clear and need to be further clarified.

5.      This paper seems to focus on the study of fatigue performance, however, the current number of test cycles of SMA elements is small compared with the actual demand, and it is difficult to explain the fatigue performance of SMA elements

Author Response

Dear Reviewer

We send one marked copy of the manuscript in which it may be seen that all the suggestions have been taken into account. The corrections are marked in the electronic version of the paper that was improved.

Reviewer #2

Point 1: The influence of temperature effect on the energy dissipation properties of SMA is well known. The reasons why Thermomechanical coupling evaluation should be further explained

Response 1: In this work, our aim is indeed to evaluate the thermomechanical coupling in SMA, however, in a much more specific context: its influence on the dynamic response of SMA (micro)cables and in their use, as structural and functional parts, and also highlighting its benefits for health monitoring. The aim of the study was clarified in the Introduction section with the following text added to page 2, line 90: “In this context, the present study proposes to contribute to the SMA cable literature by investigating the influence of heat accumulation during high loading frequency cycling on the mechanical response of NiTi SMA micro cables.”

Point 2: The effect of attenuation of vibrations is usually evaluated mainly through the hysteresis curves of energy dissipation of force and displacement, not the stress and strain curves. The main purpose of this research should be to use SMA for health monitoring instead of attenuation of vibration. The purpose of the research in this article should be further clarified

Response 2: The choice to analyze stress-strain curves instead of force-displacement resides on the possibility to accurately compare the response of different cross-sections, which was needed in our work since we compare the performance of the microcable with that of a wire. Furthermore, we have clarified to the reader the aim of the study, especially in the Introduction section, as mentioned in the response to Point 1.

Point 3: “However, in dynamic regime applications, such as vibration attenuation, where repetitive superelasticity cycles are required in loading frequencies sufficiently high, this thermal equilibrium is affected and the material experiences a gradual accumulation of heat, which consequently leads to an increase in the internal temperature. This thermomechanical coupling is the cause of the strain rate dependence observed in SMA”. The situation described in this sentence is not consistent with the requirements of the vibration attenuation

Response 3: In order to eliminate any dubiety in this statement, we have erased the specific mention to vibration attenuation. Instead, we listed some relevant applications were SMA cables can be/are applied under dynamic loadings (page 2, line 62): “These features allow the use of SMA cables as self-sensing devices, self-centering devices, damping devices, as sensors for health monitoring integrity of structures and reinforced concrete applications.”

Point 4: The research significance and conclusions of this paper are not clear and need to be further clarified

Response 4: We have remodeled our Conclusion section, now pointing out the main findings and observations in the form of a list, which is hopefully more objective. One of the main novelties of the study is the analysis of high frequency effect (up to 10 Hz) on the thermomechanical behavior of NiTi SMA microcables. Although that are some works in literature that point out the influence of strain rate on cyclic behavior of these SMA elements, none have investigated the effect of frequencies significatively higher than 10-2 Hz. Furthermore, the analysis and importance of characterizing the self-heating frequency of SMA elements used in dynamic applications is brought into attention. 

Point 5: This paper seems to focus on the study of fatigue performance, however, the current number of test cycles of SMA elements is small compared with the actual demand, and it is difficult to explain the fatigue performance of SMA elements.

Response 5: After having clarified the aim of the study (in Points 1 and 4), we would like to emphasize that the performed fatigue analysis was only preliminary. Since we have the data to perform such analysis, we believe that it brings a brief but interesting insight regarding the microcable cyclic behavior, as well as an order of magnitude to its dependence with loading frequency, and a relevant comparison with the wire used as reference element. Therefore, although we agree that the presented analysis needs to be improved to be considered an evaluation of fatigue performance, we kept the current analysis in the Results and Discussion section, however explaining to the reader the limits of the evaluation. The following text was added to page 18, line 492, to clarify this point: “We highlight that the presented fatigue study, specially concerning structural fatigue, is preliminary, since no statistical analysis was performed. Nevertheless, the results bring an order of magnitude to its dependence with loading frequency, and a relevant comparison with the wire used as reference element.”

We trust that the manuscript will meet with your approval, but should any doubt remain, please let us know.

Thank you for your attention.

Best regards,

Paulo C.S. Silva, Estephanie N.D. Grassi, Carlos J. Araújo, João M.P.Q. Delgado and António G.B. Lima

Reviewer 3 Report

1 Please provide a photo of the NiTi SMA micro cable in section 2.1.1.

2 Please provide more experimental parameters of EDS experiments in Section 2.1.2.

3 L131. Why do you choose 128 cycles?

4 L417 Please provide the meaning of Nif.

5 L459 Please unify the superscript of 1st, 2nd, and 2988th.

6 L509 Please provide the citation of the statement.

Author Response

Dear Reviewer

We send one marked copy of the manuscript in which it may be seen that all the suggestions have been taken into account. The corrections are marked in the electronic version of the paper that was improved.

Reviewer #3

Point 1: Please provide a photo of the NiTi SMA micro cable in section 2.1.1.

Response 1: We appreciate the suggestion. We have replaced the micro cable side view drawing scheme with an SEM image.

Point 2: Please provide more experimental parameters of EDS experiments in Section 2.1.2.

Response 2: We appreciate the comment. We performed new EDS tests on a sample of the SMA NiTi micro cable. This time we added more information about test parameters such as sample preparation and amplification (Section 2.1.2), and we added an image (Figure 2) to Section 3.1.1, with the results of the map (Figure 2a) and spectrum (Figure 2b).

Point 3: L131. Why do you choose 128 cycles?

Response 3: There are two main reasons to the choice of using 128 cycles in the analysis of superelastic response: a) we have chosen a 2n number of cycles because the SMA cyclic behavior evolves more intensely at the first cycles, i.e. the rate at which the stress values decrease with cycling are significantly higher at first, tending to decrease after the first cycles; b) and we have chosen 128 as a reference number of cycles to observe the differences between different frequency loadings. The following discussion was added in page 7, line 243: “It is worth noting, however, that literature on the subject shows sufficient SMA mechanical stabilization around 50 cycles [11], and even sooner if the minimal stress variation is tolerated [10,13], since the main portion of the stabilization occurs during the first cycles.”

Point 4: L417 Please provide the meaning of Nif.

Response 4: At the beginning of section 3.2.2 there is the definition of the Nif parameter. This parameter represents the number of cycles until the failure of the individual wires that constitute the micro cable.

 "Figures 15-16 show the superelastic stress vs. strain response from 1st to 7th individual failures (Nif, with i = 1 to 7)..."

Point 5: L459 Please unify the superscript of 1st, 2nd, and 2988th.

Response 5: Actually, in figures 16 and 17 the legend is 1st, 2n and xxxxth. 2n identifies the cycles 2nd, 4th, 8th, 16th, 32th, ...., 2n.

Point 6: L509 Please provide the citation of the statement.

Response 6: We appreciate the comment. We added some references.

We trust that the manuscript will meet with your approval, but should any doubt remain, please let us know.

Thank you for your attention.

Best regards,

Paulo C.S. Silva, Estephanie N.D. Grassi, Carlos J. Araújo, João M.P.Q. Delgado and António G.B. Lima

Reviewer 4 Report

Review of the manuscript by Paulo C.S. Silva et al.

NiTi SMA Superelastic Micro Cables: Thermomechanical Behavior and Fatigue Life under Dynamic Loadings

Submitted to Sensors-1944744

GENERAL COMMENTS

The paper deals with the effect of SMA thermomechanical coupling during dynamic cycling and the fatigue life of NiTi SMA micro cables. Some tests were performed on a single wire with equal cross-section area for comparison purposes. Some interesting phenomena are observed in their results, and is seems a more reliable sensor to SHM is found. According to the high-quality standards of the Sensors, the paper can be considered for publication after some revisions. Some major and minor comments are summarized as follows.

MAJOR/MINOR COMMENTS

u  The INTRODUCTION is essentially a disordered list of concise (sometimes vague and imprecise) statements about what other authors did in the past in the field. The mere sum of these statements is far from a coherent analysis of the current state of the art and certainly does not suffice to support the paper’s motivations. The authors must relate the referenced works to each other by highlighting the advancements and significant theoretical/applied results of each piece of research. The motivation of the manuscript should be illustrated more clearly and concisely.

u  As to the warming and cooling conditions, the obtained results seem not symmetry. More illustrations should be added.

u  As to the fatigue life under dynamic loadings, the cycles seem limited. I wonder why the authors just choose these number of cycles in their experiments

u  The readability of these figures should be improved. Many parametric studies are presented, and I am wonder why these types of parameters are selected and studied.

u  The experimental procedures and results should be illustrated more specifically.

u  The innovation and creativity should be illustrated more clearly in the Conclusion. It seems that some conclusions are not new, and this section is too wordy.

Author Response

Dear Reviewer

We send one marked copy of the manuscript in which it may be seen that all the suggestions have been taken into account. The corrections are marked in the electronic version of the paper that was improved.

Reviewer #4

Point 1: The INTRODUCTION is essentially a disordered list of concise (sometimes vague and imprecise) statements about what other authors did in the past in the field. The mere sum of these statements is far from a coherent analysis of the current state of the art and certainly does not suffice to support the paper’s motivations. The authors must relate the referenced works to each other by highlighting the advancements and significant theoretical/applied results of each piece of research. The motivation of the manuscript should be illustrated more clearly and concisely.

Response 1: We appreciate the suggestion and we have modified the Introduction section, especially the state-of-the art paragraph, in order to improve the coherence of our text and ideas. We also reformulated (hopefully in a clearer way) the aim of our work (in the last paragraph, page 2, line 90): “In this context, the present study proposes to contribute to the SMA cable literature by investigating the influence of heat accumulation during high loading frequency cycling on the mechanical response of NiTi SMA micro cables.”

Point 2: As to the warming and cooling conditions, the obtained results seem not symmetry. More illustrations should be added.

Response 2: Does the reviewer refer to the data in Figures 9 and 10? The temperature variations observed in the microcable and wire are actually a consequence of the release and absorption of latent heat of the reversible phase transformation, and not imposed during tests. Furthermore, it is known that this latent heat is different for forward (loading) and reverse (unloading) phase transformations, which indeed causes an asymmetry in the peak and valley temperature signals. More details on the subject can be found, for instance, on the following reference: H.M.R. de Oliveira, H. Louche, E. N. D. Grassi, D. Favier, Specific forward/reverse latent heat and martensite fraction measurement during superelastic deformation of nanostructured NiTi wires, Materials Science and Engineering: A, Volume 774, 2020, 138928, ISSN 0921-5093, https://doi.org/10.1016/j.msea.2020.138928.

Point 3: As to the fatigue life under dynamic loadings, the cycles seem limited. I wonder why the authors just choose these number of cycles in their experiments.

Response 3: If the reviewer is referring to the 128 cycles in the analysis of superelastic response under dynamic loadings, the reasons are: a) we have chosen a 2n number of cycles because the SMA cyclic behavior evolves more intensely at the first cycles, i.e. the rate at which the stress values decrease with cycling are significantly higher at first, tending to decrease after the first cycles; b) and we have chosen 128 as a reference number of cycles to observe the differences between different frequency loadings. The following discussion was added in page 7, line 243:  “It is worth noting, however, that literature on the subject shows sufficient SMA mechanical stabilization around 50 cycles [11], and even sooner if the minimal stress variation is tolerated [10,13], since the main portion of the stabilization occurs during the first cycles.”

Regarding the fatigue analysis performed at the end of the paper, the chosen variable was actually the loading frequency, which resulted in the number of cycles until the failure of the first filament (observed in Figure 20). Having clarified that, we would like to emphasize that the performed fatigue analysis was only preliminary, based on the data deriving from the dynamic analysis previously discussed in the paper. We believe that, although preliminary, the analysis brings a brief but interesting insight regarding the microcable cyclic behavior, as well as an order of magnitude to its dependence with loading frequency, and a relevant comparison with the wire used as reference element. We have clarified this to the reader, as well as the limits of the fatigue evaluation. The following text was added to page 18, line 492, to clarify this point: “We highlight that the presented fatigue study, specially concerning structural fatigue, is preliminary, since no statistical analysis was performed. Nevertheless, the results bring an order of magnitude to its dependence with loading frequency, and a relevant comparison with the wire used as reference element.”

Point 4: The readability of these figures should be improved. Many parametric studies are presented, and I am wonder why these types of parameters are selected and studied.

Response 4: We thank you for the suggestion. However, we believe that some formatting issue must have taken place when the reviewer sent his/hers comments and the mentioned figures were not specified, preventing us from improving them. As for the chosen parameters, we would like to emphasize that the main one is the loading frequency, which is relevant in the context of dynamic applications. Other previous analysis discussed in the Methodology section were used to either prepare the samples for the dynamic analysis, or to help choose other fixed experimental parameters (such as the strain range chosen based on the results in Figure 5).

Point 5: The experimental procedures and results should be illustrated more specifically.

Response 5: Although we thank the reviewer for the suggestion and believe it is a valid concern, we do not agree that the presented figures lack in specificity. Besides, the paper already counts with a total of 21 figures, which we believe are relevant, as well as we tried to make their interpretation as intuitive as we could.

Point 6: The innovation and creativity should be illustrated more clearly in the Conclusion. It seems that some conclusions are not new, and this section is too wordy.

Response 6: We have remodeled our Conclusion section, now pointing out the main findings and observations in the form of a list, which is hopefully more objective. One of the main novelties of the study is the analysis of high frequency effect (up to 10 Hz) on the thermomechanical behavior of NiTi SMA micro cables. Although that are some works in literature that point out the influence of strain rate on cyclic behavior of these SMA elements, none have investigated the effect of frequencies significatively higher than 10-2 Hz. Furthermore, the analysis and importance of characterizing the self-heating frequency of SMA elements used in dynamic applications is brought into attention. 

We trust that the manuscript will meet with your approval, but should any doubt remain, please let us know.

Thank you for your attention.

Best regards,

Paulo C.S. Silva, Estephanie N.D. Grassi, Carlos J. Araújo, João M.P.Q. Delgado and António G.B. Lima

Round 2

Reviewer 1 Report

The paper is much improved, and I suggest its acceptance.

Reviewer 2 Report

The authors have responded to the reviewers' concerns